# OpenReview forum: "Towards a Science of AI Agent Reliability"
_ICML.cc/2026/Conference — ICML 2026 regular_

### Official Review · Reviewer_7HPW · 2026-03-02

**Soundness:** 3
**Presentation:** 3
**Significance:** 2
**Originality:** 2
**Overall Recommendation:** 4
**Confidence:** 4

**Summary:**

The paper presents a framework for evaluating the reliability of AI agents, measured across four dimensions, namely consistency, robustness, predictability and safety. Frontier models from Google, Anthropic and OpenAI are evaluated using these metrics on two benchmarks, GAIA and tau. Results show that the recent models have poor performance on these metrics, despite being highly capable.

**Compliance With Llm Reviewing Policy:**

Affirmed.

**Final Justification:**

The answers provide more details (e.g., number of runs) and acknowledge the limitations. Regardless of the limitations, I believe the study makes sufficient contributions. Hence, I'm leaning towards acceptance.

**Key Questions For Authors:**

How many runs were done for metrics other than consistency?

**Limitations:**

yes

**Strengths And Weaknesses:**

Strengths:
1. The problem is well-motivated. There is a strong need to measure reliability of agents in addition to accuracy.
2. Experimental results across various models confirm that such metrics are required and current models are still quite unreliable.
3. Useful cross-domain analysis of reliability.

Weaknesses:
1. The four dimensions selected are somewhat arbitrary. While the paper provides evidence of the need for these metrics grounded in safety-critical engineering practices, it is unclear if these four dimensions are necessary or sufficient for AI agents.
2. No Related Work section discussing the evaluation of AI agent reliability in prior work. While the paper covers some related work in Section 2 and Appendix B.2, particularly around reliability evaluation in ML, important work around reliability evaluation, such as [1] are missing.
3. Only two benchmarks are evaluated and no open source models are evaluated. Particularly for safety analysis, very narrow tests are done on tau bench.
4. While the paper aims to decouple reliability metrics from capability, the current metrics are not entirely detangled. For example, robustness scores is a ratio based on the accuracy.
5. Novelty of metrics is limited. For example, prompt perturbation is often conducted in existing work. Similarly, LLM calibration is a well-studied problem.
6. No human evaluation of metrics, particularly safety and consistency.

Suggestion: Please add a Table of Contents to the Appendix because it is quite long.

[1]  Bommasani, Rishi, Percy Liang, and Tony Lee. "Holistic evaluation of language models." Annals of the New York Academy of Sciences 1525, no. 1 (2023): 140-146.

---

> ### Author Rebuttal · Authors · 2026-03-31
>
> We thank the reviewer for the detailed review and for recognizing that the problem is well-motivated and the results confirm the need for reliability metrics.
>
> ---
>
> `Concern 1: The four dimensions are "somewhat arbitrary"`
>
> **The dimensions are not arbitrary.** They are derived from a systematic survey of how safety-critical engineering disciplines decompose reliability (Section 2, Appendix B.1): consistency (repeatable behavior, DO-178C/EN 50128), robustness (graceful degradation, IEC 61508), predictability (calibrated failure awareness, nuclear PRA/aviation hazard analysis), and safety (consequence-aware risk, SIL levels). They emerge independently across multiple domains with decades of operational experience, providing stronger justification than an ad-hoc selection.
>
> Still, we do not claim this is the unique decomposition and state so explicitly. Additional dimensions (e.g., temporal degradation, multi-agent failure propagation) may be valuable as the field matures. Also, our framework is extensible: new dimensions can be added without modifying existing ones.
>
> ---
>
> `Concern 2: Missing related work (HELM)`
>
> Thanks, we have added a discussion of HELM to the paper. However, we remark that **HELM evaluates *language models* on static scenarios, whereas we focus on *AI agents***, i.e., systems taking multi-step actions with real-world consequences. This distinction is important: agent reliability introduces trajectory-level consistency, tool-use fault tolerance, action-level safety consequences (e.g., irreversible operations), and robustness to environmental perturbations beyond prompt variation. This clarifies how our agent-focused framework builds on and differs from holistic LLM evaluation.
>
> ---
>
> `Concern 3: Limited benchmarks and no open-source models`
>
> We acknowledge this limitation. Still, GAIA and $\tau$-bench were chosen for complementarity: GAIA tests multi-step reasoning with tool use in open-ended tasks; $\tau$-bench tests conversational agents in constrained customer-service domains with policy compliance requirements.
>
> On models: our focus on frontier models is deliberate, as these are most commonly deployed in real-world agent applications, and **represent the best-case scenario for reliability**. If even the most capable models exhibit the gaps we document, the problem is clearly pressing. We plan to include open-source models in the camera-ready paper version.
>
> On safety scope: we computed safety metrics on both benchmarks but found $\tau$-bench more informative due to its natural safety-relevant failure modes (incorrect transactions, policy violations). We have clarified this rationale in the paper.
>
> ---
>
> `Concern 4: Robustness metrics not decoupled from capability`
>
> **The ratio formulation is precisely what achieves the decoupling.** $R_{\text{prompt}} = \min(\text{Acc}_{\text{pert}} / \text{Acc}_0, 1)$: accuracy under perturbation *normalized by* baseline accuracy. A model at 90% dropping to 81% scores 0.9; a model at 50% dropping to 45% also scores 0.9. Both exhibit 10% relative degradation, independent of absolute capability. Without normalization, more capable models would trivially appear more robust. The ratio isolates sensitivity to perturbation from level of capability.
>
> ---
>
> `Concern 5: Limited novelty of individual metrics`
>
> We agree individual components (prompt perturbation, calibration) have been studied in isolation. Our contribution is: (1) **the unified framework**: no prior work jointly evaluates all four dimensions for AI agents, revealing insights invisible in isolation (e.g., accuracy improvements not translating to reliability across *any* dimension); (2) **adaptation to the agent setting**: extending these from static LLM tasks to multi-step trajectories with tool use, where consistency is trajectory-level, robustness includes fault-tolerance perturbations, and safety involves action consequences; (3) **the empirical finding**: the systematic capability-reliability gap across 12 frontier models and 18 months of releases is novel and practically important.
>
> ---
>
> `Concern 6: No human evaluation of metrics`
>
> We agree this is valuable future work. Our use of LLM judges follows established practice in agent evaluation; we provide full judge prompts in the appendix for reproducibility. More fundamentally, **truly reliable autonomous systems should ultimately exceed human reliability**. Decades of human factors research shows humans are also often inconsistent, poorly calibrated, and subject to systematic biases. The goal is objective, reproducible measurement, as in instrumented testing in safety-critical engineering.
>
> ---
>
> `Question: How many runs?`
>
> All metrics use the same K=5 repeated runs per task. Robustness uses these for baseline accuracy plus additional perturbation runs. Predictability aggregates across all K×T task-run pairs. Safety analyzes severity across all failing runs.
>
> We have also added a table of contents to the appendix. Thank you for this suggestion!

---

> > ### Author Rebuttal · Reviewer_7HPW · 2026-04-02
> >
> > Thank you for the detailed response. I have revised the score.

---

> > > ### Author Response · Authors · 2026-04-04
> > >
> > > We thank the reviewer for their active engagement in the rebuttal and are happy to hear that their concerns have been fully addressed!

---

### Official Review · Reviewer_7EA8 · 2026-03-10

**Soundness:** 2
**Presentation:** 2
**Significance:** 1
**Originality:** 1
**Overall Recommendation:** 2
**Confidence:** 5

**Summary:**

This paper proposes a framework for evaluating the so-called "reliability" of AI agents by decomposing it into four dimensions: consistency, robustness, predictability, and safety. The authors introduce a collection of empirical metrics for these properties and evaluate a number of frontier models on two benchmarks. The paper argues that traditional accuracy-based evaluation is insufficient and claims that the proposed framework represents a step toward a “science of AI agent reliability”.

While the topic is important, the paper suffers from several fundamental issues in problem formulation, conceptual grounding, and novelty. In particular, the proposed definition of reliability is inconsistent with established reliability engineering literature, the novelty of the evaluation methodology is limited, and the work is largely empirical without offering theoretical or methodological advances appropriate for ICML.

**Compliance With Llm Reviewing Policy:**

Affirmed.

**Final Justification:**

I appreciate the authors’ detailed response. However, I cannot raise my score unless the authors completely remove their misuse of the term “reliability” as a bundle for other properties. Reliability is not merely a generic label for a system attribute; it also refers to a well-established body of concepts, models, and methods developed by a dedicated research community, e.g., [1]. The method presented in this paper, under the label of “reliability,” has little connection to those established approaches in the reliability literature. For that reason, I am unable to raise my score unless the authors fully remove these reliability claims.

[1] Handbook of software reliability engineering, MR Lyu, 1996.

**Key Questions For Authors:**

Why it is claimed as a "science" paper when most of the work is doing empirical benchmarking?

What motivates the authors re define reliablity in the current form while reliability has already been defined in system/software engieering textbooks/standards

**Limitations:**

No.

The paper briefly acknowledges some experimental limitations (e.g., limited benchmarks and reliance on LLM judges), but the discussion is incomplete. In particular, the authors should clarify that their definition of reliability differs from established reliability frameworks in safety-critical systems literature, and discuss the limitations of benchmark-based reliability evaluation. The paper should also highlight risks of over-interpreting these metrics as evidence of real-world reliability, especially given the reliance on LLM-based judges and simulated environments.

**Strengths And Weaknesses:**

Pros:
1. An important topic.

Cons:
1. The definition of "reliability" is very concerning, which is inconsistent with established literature in system/software engieering. The central premise of the paper is a new definition of agent reliability as a combination of consistency, robustness, predictability, and safety. However, this definition does not align with established definitions of reliability in safety-critical systems literature. In traditional reliability engineering, reliability is commonly defined as: the probability that a system performs its intended function without failure for a specified period under specified conditions (According to American National Standards Institute (ANSI), or the famous Handbook of Software Reliability Engineering Edited by Michael R. Lyu ). This definition is widely used across safety-critical industries (e.g., aerospace, automotive, nuclear systems). The authors cite classical dependability literature such as Laprie’s well-known dependability framework, but the interpretation of reliability in this paper is inconsistent with that literature. In Laprie’s taxonomy, reliability is only one attribute within the broader concept of dependability, alongside availability, safety, integrity, and maintainability. The umbrella concept proposed in the paper appears much closer to dependability rather than reliability as traditionally defined. As a result, the paper introduces a new and somewhat idiosyncratic interpretation of reliability that is not clearly justified and is inconsistent with the very literature it cites. If the authors intend to redefine reliability in the context of AI agents, a much stronger conceptual justification would be required.

2. Limited novelty: aggregation of known evaluation properties. The core claim of the paper is that evaluating agents purely by accuracy is insufficient. However, this observation is already widely recognized across the AI community. Many recent benchmarks and evaluation frameworks already measure properties beyond accuracy, including robustness, calibration, safety violations, and stability under perturbations. The proposed framework essentially aggregates several existing evaluation dimensions (ie, consistency, robustness, predictability, and safety) into a single evaluation protocol. While this aggregation may be useful from an engineering perspective, it does not represent a fundamentally new conceptual contribution. It is certinaly overclaimed as a "Science" paper by just aggregating metrics...

3. Lack of theoretical contribution as claimed as a "Science" paper. The work is almost entirely empirical and benchmark-oriented. While empirical studies can be valuable, the paper makes strong claims about building a “science of AI agent reliability” but does not provide any theoretical analysis, formal framework/methodology supporting such a claim.

Revised after receving AC's comments:
While I acknowledge the AC's perspective that scientific contributions can be purely experimental, my concern lies specifically in the mismatch between the paper's ambitious positioning and its narrow scope. The paper claims to lay the groundwork for a "science of AI agent reliability"; However, a scientific approach to a problem typically requires one of two things: 1) A theoretical framework that explains why agents fail, allowing for predictions beyond the current dataset. 2) A rigorous, generalizable methodology for measuring and improving reliability that the community can adopt. This paper provides neither IMHO. It presents a series of empirical observations on specific benchmarks. While these observations are interesting, they remain a collection of data points rather than a structured path toward a "science" of "reliability". To substantiate the claim of building a science, the paper would need to synthesize its empirical results into a reusable methodology or a taxonomy of failure modes that helps researchers categorize and predict issues in new, unseen agents. Currently, the work is descriptive (benchmark results) rather than prescriptive or foundational.

4. The central research question is not well-motivated. The paper frames its core research question as “How should we define and evaluate agent reliability?” However, reliability is already a well-established concept with widely accepted definitions in system/software engineering research. The real challenge in AI is typically how to ensure or guarantee reliability, not how to define it again.

5. Minor Issues:
a. Some metrics rely on LLM-based judging, which introduces additional uncertainty and evaluation bias.
b. The experiments rely on only two benchmarks, limiting the generality of the conclusions.
c. The aggregation of multiple heterogeneous metrics into a reliability score is not theoretically justified.

---

> ### Author Rebuttal · Authors · 2026-03-31
>
> We thank the reviewer for engaging with our work and for raising questions about the relationship to classical reliability engineering.
>
> ---
>
> `Concern 1: Relationship to established reliability definitions`
>
> The reviewer correctly identifies a tension between the classical ANSI definition ("probability that a system performs its intended function without failure for a specified period under specified conditions") and our broader usage. We agree this deserves explicit discussion, and have added a paragraph addressing it.
>
> The core issue is that **AI agents do not fit the assumptions behind the ANSI definition**: they are stochastic, operate in open-ended environments, and lack a fixed "specified period" or "specified conditions." The standards we cite have already evolved beyond this narrow formulation in practice: IEC 61508 defines safety integrity levels based on fault tolerance, diagnostic coverage, and systematic capability; DO-178C requires consistency testing and robustness to environmental conditions. Our work follows this same trajectory, *operationalizing* reliability for a new class of systems.
>
> On "dependability": we considered this carefully. Laprie's framework encompasses availability and maintainability, which are infrastructure-level properties less applicable to individual agent evaluation. We have added a paragraph explicitly discussing this terminological choice and the trade-offs involved.
>
> ---
>
> `Concern 2: Novelty of the framework`
>
> We agree that individual evaluation dimensions (robustness, calibration, etc.) have been studied before, and **we do not claim novelty for any single metric**. Our contribution is: (1) adapting these concepts to *agents*, where they take qualitatively different forms (trajectory-level consistency, tool-fault robustness, action-consequence safety); (2) providing a unified framework enabling systematic comparison across all dimensions; (3) empirical findings showing the capability-reliability gap is systematic across 12 frontier models and 18 months of releases. Studying these dimensions jointly reveals insights invisible in isolation, e.g., that high robustness does not predict good calibration. We would welcome pointers to existing frameworks that jointly evaluate these dimensions for AI agents, as our literature survey (Section 2, Appendix B.2) did not find any.
>
> ---
>
> `Concern 3: "Science" framing`
>
> We want to clarify our intent. The title "*Towards* a Science" is deliberately aspirational. The reviewer's revised comment helpfully distinguishes between (1) a theoretical framework and (2) a rigorous, generalizable methodology. **We aim to provide (2)**: a structured decomposition with 12 computable metrics, grounded in safety-critical engineering, applied across models and benchmarks. We see this as measurement infrastructure that future theoretical work can build on. If the title overpromises, we are open to alternatives such as "Measuring AI Agent Reliability" or "A Framework for Evaluating AI Agent Reliability."
>
> We also agree that connecting empirical observations to explanatory frameworks would strengthen the field. The LLM agent field is inherently empirical at this stage: failure modes are not yet taxonomized and the link between training and deployment reliability is not understood. Our finding that capability gains do not translate to reliability improvements is precisely the kind of empirical regularity that invites and constrains future theoretical explanation.
>
> ---
>
> `Concern 4: Motivation for the research question`
>
> The reviewer makes the point that reliability is well-established in systems engineering. We agree. However, our observation is that **this knowledge has not yet been adapted to AI agents**, which differ from traditional systems in fundamental ways: stochastic behavior, emergent failure modes, open-ended operating conditions. To illustrate: the ANSI definition requires "specified conditions," but what are these when the agent's environment includes arbitrary web pages, APIs, and user interactions? The reviewer's framing that "the real challenge is how to ensure reliability" is one we share, and our framework addresses the prerequisite: *measuring* reliability so that improvement efforts have a clear target to optimize for. This is a common progression in engineering: define metrics, establish baselines, then improve.
>
> ---
>
> `Concern 5: Minor issues`
>
> **LLM-based judging:** We agree this introduces uncertainty and give full judge prompts in the appendix for transparency. **Two benchmarks:** This is a stated limitation; see discussion with reviewer 7HPW. **Aggregation:** We pick uniform averaging as neutral default but alternative weightings are possible.
>
> ---
>
> `Limitations`
>
> We note that the paper contains an explicit Limitations paragraph discussing benchmark scope, LLM judge reliance, metric subjectivity, and the benchmark-to-deployment gap. We are happy to expand this with the gaps identified by the reviewer in the camera ready paper.

---

> > ### Author Rebuttal · Reviewer_7EA8 · 2026-04-02
> >
> > 1. On the relationship to established reliability definitions
> >
> > The rebuttal argues that classical reliability definitions do not transfer cleanly because AI agents are stochastic, open-ended, and lack a fixed “specified period” or “specified conditions”; I agree that AI agents are different from traditional software in important ways. However, this does not weaken the relevance of classical reliability concepts; rather, it makes them more necessary to adapt carefully and explicitly. Note, software in 80s/90s was used in the same way as AI agent now without those caveats.
> >
> > In my view, the right response to open-ended agent settings is not to bypass notions such as specified conditions, operational envelope, failure definition, demand/exposure model, and operational profile, but to develop modern equivalents for them. Indeed, the paper itself repeatedly motivates its framework through safety-critical engineering examples that presuppose precisely these kinds of concepts: repeated execution under the same conditions, degradation across the operational envelope, explicit failure probabilities, and bounded worst-case consequences.
> >
> > So my concern is not that the authors acknowledge agents are different. My concern is that the manuscript jumps too quickly from “agents are different” to “therefore reliability should be redefined as consistency + robustness + predictability + safety” without a sufficiently rigorous acdemic argument.
> >
> > 2. The appeal to safety-critical standards actually strengthens my concern!
> >
> > The rebuttal cites IEC 61508 and DO-178C as evidence that practice has already evolved beyond a narrow textbook definition. I do not think this supports the paper’s position in the way intended. These standards are not examples of abandoning reliability concepts; they are examples of operationalizing them more rigorously through explicit assumptions about system safety/reliability engieering concpets (way before Ai agent exisits): hazardous failures, fault tolerance, coverage, assurance activities, opertional profiles.
> >
> > Likewise, in the paper itself, Table 1 invokes exemplars such as SIL 4 dangerous failure probability targets and aviation catastrophic failure targets, while the surrounding discussion emphasizes nominal conditions, operational envelopes, failure probabilities, and safe modes. These are not loose analogies; they are precisely the kinds of system-engineering constructs that make reliability claims meaningful.
> >
> > For that reason, I remain unconvinced that one can invoke safety-critical engineering as conceptual grounding while simultaneously relaxing the need for explicit definitions of operating conditions, failure semantics, and exposure assumptions for AI agents. If anything, the authors’ own examples suggest the opposite: if agent reliability is to be treated seriously, the field should work harder on formalizing those concepts for AI agents, not sidestep them.
> >
> > 3. I remain unconvinced that the paper is defining reliability rather than assembling a useful evaluation bundle
> >
> > I agree that jointly measuring some behavioral dimensions may be useful. However, that still does not resolve the core conceptual issue. Consistency, robustness, predictability, and safety are all good properties to evaluate. But it does not follow that their aggregation constitutes “reliability” in the classical or adapted engineering sense. The paper itself operationalizes these as separate metrics, then aggregates several of them through uniform averaging, while excluding safety from the overall aggregate because safety is treated as a tail-risk phenomenon. That design may be pragmatically defensible, but it reinforces my impression that this is an engineering dashboard rather than a conceptually grounded redefinition.
> >
> > In other words, I do not object to the framework being useful. I object to the manuscript presenting this framework as though it has already solved the deeper conceptual question of what “reliability” should mean for AI agents. I am open to change my score if the authors can reframe the paper for other terms like trustworhyness, etc.
> >
> >
> >
> >
> > *****comments on the 2nd round resp.:****
> >
> > I appreciate the authors’ detailed response. However, I cannot raise my score unless the authors completely remove their misuse of the term “reliability” as a bundle for other properties. Reliability is not merely a generic label for a system attribute; it also refers to a well-established body of concepts, models, and methods developed by a dedicated research community, e.g., [1]. The method presented in this paper, under the label of “reliability,” has little connection to those established approaches in the reliability literature. For that reason, I am unable to raise my score unless the authors fully remove these reliability claims.
> >
> > [1] Handbook of software reliability engineering, MR Lyu, 1996.

---

> > > ### Author Response · Authors · 2026-04-04
> > >
> > > We thank the reviewer for their detailed response. This exchange has been productive, and we believe we are closer to agreement than the remaining disagreements might suggest.
> > >
> > > ---
> > >
> > > `On the relationship between our framework and classical reliability engineering`
> > >
> > > **We agree with the reviewer's core thesis.** The right path forward is to develop modern equivalents of operating conditions, failure semantics, and exposure models for AI agents. We do not advocate bypassing these concepts.
> > >
> > > Where we differ is on *sequencing*, not on *direction*. Formalizing operating conditions for agents is genuinely hard: what constitutes the "operational envelope" of an agent that can browse arbitrary web pages, call arbitrary APIs, and encounter inputs never specified at design time? Our paper provides measurement tools that the field can use *now*, while this formalization proceeds. Measurement and formalization can develop in parallel, each informing the other. Based on this feedback, we have added a discussion to Section 2 identifying these open formalization challenges and framing our framework as a complement to, not a substitute for, this work.
> > >
> > > The reviewer's analogy to 1980s/90s software supports this: reliability engineering for software *did* begin with empirical measurement (defect counting, failure rate estimation) before formal frameworks matured. For example, *Musa's software reliability models* were built on observation first, then formalized. We see our work as playing a similar role.
> > >
> > > ---
> > >
> > > `On "defining reliability" vs. "assembling a useful evaluation bundle"`
> > >
> > > This is the crux, and we think that the gap is narrower than it appears. **We are not claiming to have solved what reliability should mean for AI agents.** We propose a *working operationalization* that enables systematic measurement. We agree this is not the same as a formal definition with axiomatic foundations.
> > >
> > > We acknowledge that our original phrasing "decomposing reliability into four dimensions" may have read as a definitional claim rather than an empirical synthesis, and we have revised this language. **We are not proposing these dimensions from scratch, nor copying a single standard.** We surveyed aviation, nuclear, automotive, and railway engineering and found that the *same* underlying concerns recur independently, though under different names: "deterministic execution" (DO-178C) and "repeatable response" (NRC) both map to consistency; "fault tolerance" (IEC 61508) and "graceful degradation" (EN 50126) both map to robustness. This cross-domain synthesis is one of our contributions: identifying convergent structure across independent traditions, then adapting it to agents with computable metrics. The reviewer acknowledges these are "all good properties to evaluate"; our contribution is showing *why* these four emerge as the recurring concerns. To make this derivation transparent, we have added an appendix section documenting, standard by standard, which reliability concerns each domain emphasizes and how our dimensions map to them. This makes the synthesis auditable.
> > >
> > > We have revised the paper as follows: (1) revised the title to "Measuring AI Agent Reliability"; (2) added language stating this is a working operationalization, not a formal redefinition; (3) added a discussion of open formalization challenges as future work. These changes reflect the reviewer's feedback.
> > >
> > > ---
> > >
> > > `On the dependability discussion`
> > >
> > > We agree this was insufficiently developed. We have added a discussion mapping Laprie's dependability attributes to the agent setting, explaining which we address, which we exclude, and why (e.g., availability and maintainability concern infrastructure-level operations rather than agent behavioral properties).
> > >
> > > ---
> > >
> > > `On the suggestion to reframe as "trustworthiness"`
> > >
> > > We considered this carefully. However, "trustworthiness" in the AI literature encompasses ethical and social dimensions: fairness, transparency, accountability, privacy (see NIST AI RMF, EU AI Act). Our paper does not address these, and adopting the term would invite criticism about missing dimensions. **"Reliability" remains the more precise term for what we measure.** The reviewer's concern, as we understand it, is not the word itself but the strength of the claim. The title change and reframing as a working operationalization address this directly without a terminological mismatch.
> > >
> > > ---
> > >
> > > `On common ground`
> > >
> > > We think we are in agreement with the reviewer that: (1) agent reliability is important; (2) classical concepts should inform the approach; (3) the four dimensions are "good properties to evaluate"; (4) deeper formalization is needed long-term. The remaining disagreement is about *what level of formalization is needed before empirical measurement constitutes a contribution*. We believe the revisions above demonstrate our commitment to the positioning the reviewer calls for, and we hope the reviewer will re-consider our contribution in light of these clarifications.

---

### Official Review · Reviewer_HY5n · 2026-03-13

**Soundness:** 3
**Presentation:** 3
**Significance:** 4
**Originality:** 4
**Overall Recommendation:** 5
**Confidence:** 4

**Summary:**

The paper argues that accuracy alone is insufficient to measure AI agents and proposes a reliability evaluation framework with 12 metrics spanning four dimensions: consistency, robustness, predictability, and safety. Empirical findings indicate that recent gains in model capability have yielded minimal improvements in reliability, highlighting a systematic gap between capability and reliability.

**Compliance With Llm Reviewing Policy:**

Affirmed.

**Final Justification:**

This is a solid and valuable paper that goes beyond accuracy to systematically evaluate agent reliability. I had some initial concerns about the concepts and metric design, but the rebuttal addressed them clearly. Overall, the work is rich in content (with detailed appendix) and makes a meaningful contribution.

**Key Questions For Authors:**

no further questions

**Limitations:**

Yes, the authors adequately discussed the limitations and potential negative societal impact of their work.

**Strengths And Weaknesses:**

The paper addresses a timely and important gap in AI agent evaluation. The shift from accuracy-centric measurement toward multi-dimensional reliability is well-motivated, particularly given the growing deployment of agents in safety-critical settings. The benchmarking approach is methodologically sound, with multi-run evaluation and perturbation testing lending credibility to the empirical claims.

**Concerns**
1. The paper draws a distinction between reliability and capability, with claims such as "reliability gains lag behind capability progress." However, the concepts and metrics proposed for reliability appear to be capability-dependent constructs. Also, my understanding is that "capability" in Figure 1 refers to accuracy. While the phrasing is eye-catching and easy to understand, the distinction reads more as a rhetorical contrast than a principled conceptual separation.

2. The paper adopts "reliability" as an umbrella term that is decomposed into consistency, robustness, predictability, and safety, while the conceptual coherence of this grouping might not be fully established. For example the inconsistency in trajectory does not necessarily indicate an unreliable system -- variance can be acceptable or even expected under certain conditions.
The paper does note that "the choice of specific metrics within each reliability dimension involves subjective decisions," which is quite fair, but this might make it more important to justify why "reliability" is the most appropriate umbrella, rather than alternatives like "dependability" or "trustworthiness," or any other possible terms. As it stands, the framework reads somewhat like a collection of related metrics organized around a compelling narrative.

3. Some metric formulations would benefit from additional theoretical grounding. As one example, the harm score  S_harm applies an exponential transformation to severity scores assigned by an LLM judge. While exponential scaling is common in safety engineering where harms are treated as orders of magnitude differences, it is worth clarifying what assumptions are being made about the LLM judge's output scale. For instance, are the severity ratings assigned by the LLM judge assumed to be linear in magnitude, or can any scale be applied? More broadly, the proposed formulations would benefit from clearer motivation and guidance on the conditions under which each is appropriate.

---

> ### Author Rebuttal · Authors · 2026-03-31
>
> We thank the reviewer for the thoughtful review and for recognizing the timeliness, methodological soundness, and significance of our work.
>
> ---
>
> `Concern 1: Reliability vs. capability distinction`
>
> **The distinction is principled, not merely rhetorical.** Two agents at the same accuracy can have fundamentally different reliability profiles: one fails on a fixed, identifiable subset of tasks (enabling targeted debugging and safe deployment with guardrails), while the other fails unpredictably on a different subset each run. Robustness further illustrates this: a model can achieve high accuracy yet degrade sharply under minor input variation, a failure mode that accuracy alone cannot reveal. Predictability provides another example: two models at identical accuracy can differ dramatically in calibration, with one accurately signaling when it is likely to fail and the other expressing high confidence regardless. These distinctions are invisible to accuracy metrics yet critical for deployment decisions.
>
> The distinction is vacuous only at 100% accuracy. Current frontier models are far from this regime (the best models achieved below 70% on GAIA), and as the field designs harder benchmarks, this is likely to persist. Our empirical results demonstrate directly that models with substantially higher accuracy do not show universal reliability improvements across *any* of the four dimensions.
>
> **On Figure 1:** We have clarified the caption title to state that we use task accuracy as the capability measure. The figure already plots accuracy directly and does not introduce ambiguity. The claim is that the field's focus on accuracy improvements have not translated into reliability improvements, which our data supports.
>
> ---
>
> `Concern 2: Conceptual coherence of "reliability" as umbrella term`
>
> Our choice is grounded in safety-critical engineering practice, not arbitrary. We draw on aviation (DO-178C), nuclear power (IEC 61513), automotive (ISO 26262), and railway systems (EN 50126), where reliability is the established term for the multi-dimensional property we study. Our four dimensions map to concerns these domains identify as constitutive of reliable system behavior: consistency (repeatable behavior under nominal conditions), robustness (graceful degradation under stress), predictability (calibrated failure awareness and bounded failure modes), and safety (consequence-aware risk assessment).
>
> We prefer "reliability" over "dependability" because the latter (per Laprie, as cited in our work) encompasses availability and maintainability, which are infrastructure-level concerns (uptime, mean time to repair) less applicable to individual agent evaluation. "Trustworthiness" is an even broader term, often including ethical and social dimensions (fairness, transparency, accountability) outside our scope. **"Reliability" most precisely captures the operational behavioral properties we measure.** We have added a paragraph to the paper explicitly justifying this terminological choice, highlighting the contrast with these alternatives.
>
> **On trajectory consistency:** We agree that variance can sometimes be acceptable or even desirable. We now note explicitly in the paper that trajectory consistency matters most in domains demanding auditability or process reproducibility, i.e., settings where stakeholders must verify not just *what* the agent concluded but *how* it got there. It matters less in open-ended or creative tasks where diverse solution paths are valued. More broadly, our framework is designed to be modular: practitioners can weight, exclude, or reinterpret dimensions relative to their specific deployment context, rather than treating all dimensions as universally required.
>
> ---
>
> `Concern 3: Theoretical grounding of harm scoring `
>
> The LLM judge does not output scores on an arbitrary scale. We define a severity scale with explicit anchors (see Appendix D.3.5 for details): 0–1 (informational), 1.5–3 (low), 3.5–5.5 (medium), 6–8 (high), 8.5–10 (critical), with contextual instructions guiding the judge (e.g., whether the action was requested, whether it involved test data, whether the agent recovered). The exponential transformation maps mean severity to a [0, 1] reliability score. Because we control the scale definition, the transformation operates on a known, anchored input range. The assumptions are ones we set, not ones we inherit from the judge.
>
> Severity assessment is inherently subjective, a challenge shared by all safety evaluation, including traditional safety-critical engineering (e.g., SIL level assignment under IEC 61508 or hazard severity categories in aviation). **The key contribution is making this assessment *explicit and systematic* rather than leaving it implicit.** By defining clear anchors and using a standardized evaluation protocol, we enable reproducible comparisons across agents, even if the absolute scale is debatable. We have added a brief discussion of this point to strengthen the motivation.

---

> > ### Author Rebuttal · Reviewer_HY5n · 2026-04-01
> >
> > Thank you for the detailed rebuttal. All my concerns are resolved and I am happy to increase my score accordingly.

---

> > > ### Author Response · Authors · 2026-04-01
> > >
> > > We thank the reviewer for their swift response and are happy to hear that their concerns have been fully addressed! Our work has already improved thanks to their feedback.

---

### Decision · Program_Chairs · 2026-04-30

**Decision:**

Accept (regular)

**Comment:**

I am conflicted on how to handle this paper. The authors have chosen an important topic, and it seems they have made some progress towards offering people helpful ways to think about this topic. This is reflected in the positive evaluations of 2 reviewers.

On the other hand, reviewer 7EA8 raises some credible concerns that the authors have moved into territory where they must be more careful with their language and framing. Software reliability is a serious field with years of academic and practical work carefully defining concepts and metrics. These terms may or may not be relevant to how we are using large language models, but they exist and it is important to situate new work with respect to these existing concepts.

I do think that the authors and reviewer 7EA8 had a helpful exchange that narrowed their differences, but reviewer 7EA8 still has the position that their concerns have not been fully heard. To do this would, I think, require a section of the paper talking about how the contributions of this paper intersect established concepts.

The difficulty in assessing this paper boils down to weighing the positive reactions from two reviewers who see the work as worthy of publication despite 7EA8's concerns, with the risk of confusion caused by the possibility introducing the concepts in this paper without full discussion of the relationship to established software reliability concepts. If the paper is accepted (and, frankly even if it's not), the authors really need to find a way to fit this discussion into the paper - wherever it ultimately winds up getting published.